# Efficacy of Single Tocilizumab Administration in an 88-Year-Old Patient with Severe COVID-19 and a Mini Literature Review

**DOI:** 10.3390/geriatrics7010022

**Published:** 2022-02-21

**Authors:** Cid Ould Ouali, Nadia Ladjouzi, Khidher Tamas, Hendriniaina Raveloson, Jihene Ben Hassen, Nesrine El Omeiri, Georges Zouloumis, Mohamed Moataz Al Zoabi, Muneer Asadi, Aziza Jhouri, Joël Schlatter

**Affiliations:** 1Department of Post-Acute and Rehabilitation Care, Department of Critical Geriatric Medicine, Hospital of Paul Doumer, Assistance Publique des Hôpitaux de Paris (AP-HP), 60140 Labruyère, France; cid.ouldouali@aphp.fr (C.O.O.); nadia.ladjouzi@aphp.fr (N.L.); khidher.tamas@aphp.fr (K.T.); hendriniaina.raveloson@aphp.fr (H.R.); jihene.benhassen@aphp.fr (J.B.H.); nesrine.elomeiri@aphp.fr (N.E.O.); georges.zouloumis@aphp.fr (G.Z.); mohamedmoataz.alzoabi@aphp.fr (M.M.A.Z.); muneer.asadi@aphp.fr (M.A.); 2Pharmacy, Hospital of Paul Doumer, Assistance Publique des Hôpitaux de Paris (AP-HP), 60140 Labruyère, France; aziza.jhoury@aphp.fr

**Keywords:** COVID-19, Tocilizumab, case report, elderly, inflammatory syndrome

## Abstract

The new coronavirus disease 2019 (COVID-19) could be associated with elevated inflammatory cytokine levels, suggesting the involvement of cytokine release syndrome. This syndrome is characterized by release of interleukin 6 correlated with COVID-19 severity and mortality. Targeting IL-6 with Tocilizumab treatment could be a potential therapeutic option for old patients. We report the case of an 88-year-old man with COVID-19 disease who presented at the admission with anemia, fever, oxygen desaturation (92%), and inflammatory syndrome (C-reactive protein (CRP) at 182.5 mg/L; reference range <5.0 mg/L). After remaining CRP level increase (206.6 mg/L), Tocilizumab administration led to rapid clinical outcome and resolution of his inflammatory syndrome. This case report represents a supplementary data confirming the efficacy and safety of Tocilizumab for COVID-19 in elderly patients.

## 1. Introduction

The new coronavirus pneumonia, first identified in the province of Hubei (China) and designated by the World Health Organization (WHO) as coronavirus disease 2019 (COVID-19), induced a viral infection by severe acute respiratory syndrome coronavirus 2 (SARS-CoV-2) that affected several organs involving the respiratory, renal, cardiovascular, central nervous, and gastrointestinal systems [1,2]. This multi organ dysfunction syndrome could be explained by the immune dysregulation caused by the SARS-CoV-2, resulting in hyperinflammation on the immune systems and cytokine release syndrome (cytokine storm) [3,4]. In this context, elderly patients are more likely to experience a more severe course of disease due to alterations in both the innate and adaptative immune systems [5]. In advanced age, it has been reported that an increase in the levels of circulating pro-inflammatory cytokines such as interleukins, growth factors, and proteases [6,7]. These elements could contribute to develop severe disease in elderly adults over 65 years old that represent 80% of the hospitalizations with a higher risk of death than those under 65 [5]. Older adults with severe COVID-19 could rapidly progress into a cytokine storm that involves hyperactivation of the immune system and hypercoagulation in the small blood vessels [8].

Because COVID-19 severity and mortality are correlated with the high levels of interleukin-6 (IL-6), administration of Tocilizumab, a recombinant monoclonal antibody which binds specifically to IL-6 receptors, was used with a decrease in fever, lower oxygen requirements, and a reduction in mortality [9,10,11,12]. However, few data exist on the effectiveness and safety of the drug treatment in elderly patients [13,14].

In this case, we describe the successful treatment by single Tocilizumab administration of an 88-year-old patient with severe COVID-19.

## 2. Case

An 88-year-old man was admitted in our geriatric rehabilitation unit after a blood transfusion for anemia at 5 g/L. The past medical history of the patient included epilepsy, ischemic heart failure treated, arterial hypertension, dyslipidemia, and obesity. At baseline, he had partial deafness and was brought to medical attention due to loss of autonomy for personal hygiene and dressing. After the admission, asymptomatic SARS-CoV-2 infection was confirmed by reverse transcription polymerase chain reaction (RT-PCR) test and the patient was transferred to our COVID-19 unit. He had sustained fever (38 °C), heart rate at 100/min, blood pressure at 142/65 mmHg, and the peripheral capillary oxygen saturation (SpO2) rose to 92% by 2 L/min oxygen. At the same time, blood sample analysis found elevated levels of C-reactive-protein (CRP 182.5 mg/L) a high marker of inflammation, an anemia (hemoglobin 10.8 g/dL), hyponatremia (132 mmol/L), hyperkalemia (4.53 mmol/L), and hyperuricemia (11.8 mmol/L). The patient received paracetamol (3 g/day), ramipril (5 mg/day), and lansoprazole (15 mg/day). Table 1 shows the main laboratory values made to date. 

On day 2, he presented a sudden oxygen desaturation (SpO2 90%) stabilized with the administration of oxygen at high flow (4 L/min). The thoracic radiography showed an alveolar and interstitial diffuse damage and a left basal outpouring (Figure 1). 

At this time, the patient received dexamethasone (6 mg daily for 10 days), enoxaparin (4000 UI daily for 30 days), and combination of amoxicillin and clavulanic acid (1 g/200 mg × 3/day for 2 days) as the recommendations for the therapeutic management of COVID patients from the Paris University Hospitals (AP-HP) and the European Respiratory Society guideline. On day 3, a computed tomography angiogram of the chest was ordered and revealed left segmental pulmonary emboli without sign of pulmonary arterial hypertension (Figure 2).

On day 4, spiramicin (3 MUI × 3/day) was added and ceftriaxone 1 g daily to treat a urinary tract infection to *Morganella morgana*. The combination of amoxicillin and clavulanic acid was stopped without impact on the fever. By hospital day 8, its clinical state was judged to be critical with persistent high fever (38.9 °C), blood pressure at 147/59 mmHg, heart rate 86/min, oxygen saturation 90% with 5 L/min, and high level of CRP 206.6 mg/L. Figure 3 summarizes major clinical parameters and CRP levels obtained during the hospital course.

At this time, the choice of Tocilizumab treatment was discussed and validated in a multidisciplinary meeting with geriatricians and clinical pharmacists.

The patient received a single dose of intravenous Tocilizumab at 8 mg/kg (600 mg) with no clinical improvement. The same day, the clinical state of the patient progressed favorably with normal temperature at 36.3 °C, heart rate 74/min, and blood pressure 159/71 mmHg. The CRP level was significantly decreased to 23.3 mg/L after 5 days of drug administration and normalized at day 10 of drug administration. The oxygen saturation was increased more progressively, and the oxygen discharge was effective by day 28 after the Tocilizumab administration. Four weeks later, the patient was seen in an out-patient setting and had been doing well.

## 3. Discussion 

We report the case of an 88-year-old man with risk factors including an older age that was treated successfully by Tocilizumab for a severe COVID-19 disease. Clinical data showed that the patient’s vital parameters improved after drug treatment. Moreover, a significant decrease in CRP and fever levels was observed just after a single infusion, suggesting the potential impact of Tocilizumab on the cytokine release syndrome. Therefore, our patient did not experience any adverse events.

Because studies in severe COVID-19 observed the association with elevated levels of IL-6 and mortality, randomized clinical trials suggested benefit of the anti-IL-6 receptor monoclonal antibody Tocilizumab with reported heterogenous results [15,16,17,18,19]. In a randomized clinical trial that included 130 patients hospitalized with COVID-19 and moderate-to-severe pneumonia, Tocilizumab did not reduce the mortality over 28 days in comparison with the usual care group [16]. These results were supported by the study by Tsai et al. on the impact of Tocilizumab administration on mortality in severe COVID-19 that recorded no statistical difference between the Tocilizumab group and the no Tocilizumab group (odds ratio, 1.0; 95% confidence interval, 0.465–2.151; *p* = 1.00) [20]. In a recent research article, a trial that examined survival varied with baseline CRP levels demonstrated the administration of Tocilizumab was beneficial in patients with CRP levels greater than 15.0 g/L [21]. Some studies have also evaluated the Tocilizumab treatment in patients with severe COVID-19, suggesting its efficacy on mortality [22,23,24]. The TCZ efficacy on mortality on day 14 and 28 was retrospectively evaluated in 62 patients (57.4 ± 14.3 years) who received one to two doses of TCZ (400–800 mg every 12 h) [22]. There was a significant difference in the 28-day (TCZ = 62 and control = 86) and in the 14-day (9.7% vs. 24.4%, *p* = 0.022) fatality rate among the TCZ-treated patients and control group. The inflammatory marker CRP decreased in the TCZ-treated patients, indicating a response to the IL-6 blocker. Moreover, a meta-analysis including 10 studies involving 1675 severe COVID-19 patients (median age >52 years) revealed a significant difference in mortality in the Tocilizumab group compared to the control group (19.5% vs. 28.3%, *p* < 0.00001) [24]. As in our clinical case, a Spanish cohort of elderly COVID-19 patients (mean age 85.2 years) found benefit in patients receiving Tocilizumab and corticosteroids (adjusted estimation HR 0.09, 95% CI: 0.01–0.74) [23]. Elderly patients are frequently absent from clinical trials even though we need efficacy and safety data in this population. More information is needed regarding the dose schedule of Tocilizumab treatment, eligibility criteria to select the patients, and association with other therapeutic options such as corticosteroids or antiviral drugs.

## 4. Conclusions

In conclusion, Tocilizumab may be an optional treatment in elderly patients with severe COVID-19 associated with high CRP level. However, further research is needed to identify optimal elderly patient population for treatment with Tocilizumab, and to confirm its effectiveness and safety.

## Figures and Tables

**Figure 1 geriatrics-07-00022-f001:**
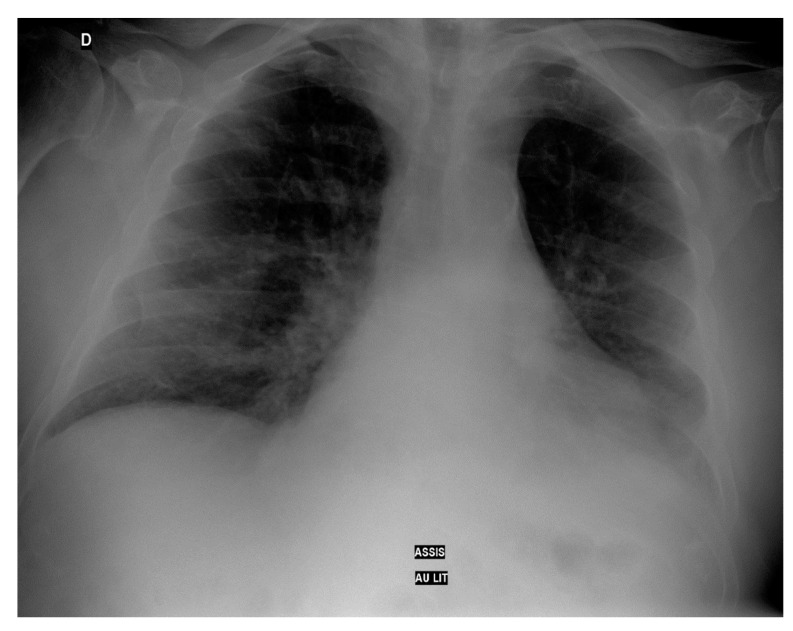
Thoracic radiography at day 2.

**Figure 2 geriatrics-07-00022-f002:**
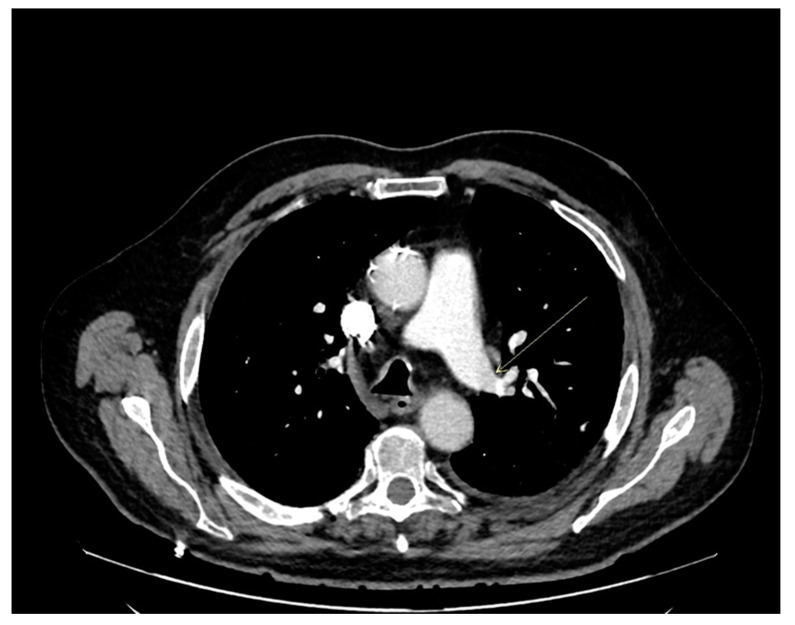
Computed tomography angiogram of the chest at day 3. (arrow shows the emblism).

**Figure 3 geriatrics-07-00022-f003:**
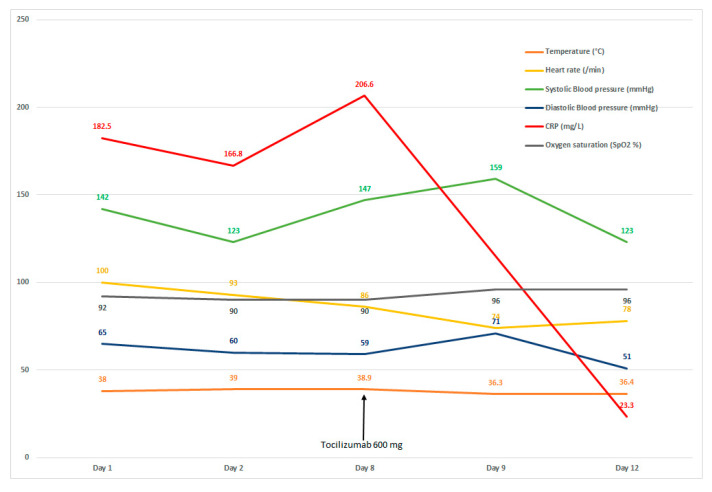
Clinical parameters from day 1 to day 12 (Tocilizumab administration at day 8).

**Table 1 geriatrics-07-00022-t001:** Main laboratory values made to date.

Parameters (Laboratory Reference)	Admission	Day 2	Day 8	Day 13
CRP (<5.0 mg/L)	182.5	176.8	206.6	23.3
Hemoglobin (12.9–16.7 g/dL)	10.8	10.7	9.9	10.0
Natremia (136–145 mmol/L)	132	135	137	139
Kalemia (3.40–4.50 mmol/L)	4.53	4.56	4.38	3.75
Lymphocytes (1.070–4.100 g/L),	1.470	1.010	0.690	0.48
Blood creatinine (62.0–106.0 µmol/L)	99.9	103.3	78.4	88.1
Blood urea (2.86–8.21 mmol/L)	11.8	12.9	8.2	7.1
MDRD ^1^ (>90)	62	60	82	72
NT-proBNP ^2^ (50.0–125.0 pg/mL)		-	9118.0	3819.0

^1^ MDRD: modification of diet in renal disease ^2^ NT-proBNP: pro-brain natriuretic peptide.

## Data Availability

The data presented in this study can be requested to the corresponding author. The data are not publicly available due to confidentiality and anonymity.

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
