# Peer review of "Efficacy of Single Tocilizumab Administration in an 88-Year-Old Patient with Severe COVID-19 and a Mini Literature Review"

_geriatrics, 2022, doi:10.3390/geriatrics7010022_

Round 1

Reviewer 1 Report

Ouali and colleagues present a case report manuscript in which they communicate the results obtained after administrating a single dose of tocilizumab (8mg/kg) in an 88-years-old patient with severe COVID-19. They report that this tocilizumab administration led to rapid resolution of the patient’s inflammatory syndrome.

The manuscript provides sufficient background, correctly describes the clinical parameters of the case and the results obtained after the tocilizumab administration. However, the reviewer considers that Figure 1 can be improved by adding, for example, a vertical line indicating the moment of the tocilizumab administration. The reviewer recommends the authors use a more specific statistical software like GraphPad or Stata, among others.

The results communicated in this manuscript support the hypothesis that blocking IL-6 could be useful in reducing COVID-19 severity and mortality. Moreover, these results are in concordance with RECOVERY and REDMAR-CAP trials indicating that tocilizumab reduces mortality in severe COVID-19 patients. Although in their work there is only one patient included, the reviewer considers the manuscript valid as it generates evidence of the effectiveness of tocilizumab treatment in elderly severe-COVID-19 patients. This data could be useful in clinical practice and COVID-19 patients management.

Author Response

Dear reviewer,

  1. The title was changed as recommended by reviewer.
  2. Tocilizumab has been reported with capital letter in the text.
  3. Table 1 with laboratory values made to date was added.
  4. Figure 1 represented the thoracic radiography was added.
  5. The discussion was divided from the conclusion.
  6. The discussion was extended reporting other articles as suggested by reviewer.

Reviewer 2 Report

In this manuscript, the authors demonstrated that tocilizumab treated the elderly patient with COVID-19. Every case report is valuable, and the comparison with similar cases (tocilizumab treatment for the elderly (over 70 or 80) with COVID-19) in discussion will be helpful to improve the manuscript.

Author Response

Dear reviewer,

  1. The title was changed as recommended by other reviewer.
  2. Tocilizumab has been reported with capital letter in the text.
  3. Table 1 with laboratory values made to date was added.
  4. Figure 1 represented the thoracic radiography was added.
  5. The discussion was divided from  the conclusion.
  6.  The discussion was extended reporting other articles as suggested by other reviewer.

Reviewer 3 Report

the effect of tocilizumab on CRP reduction is already known.

concomitant antibiotic therapy influenced patient outcome.

After computed tomography angiogram of the chest was ordered and revealed left segmental pulmonary emboli, why dose of LMWH is not adjusted for therapeutic treatment?

Author Response

(The authors gave the same response as above.)

Reviewer 4 Report

The manuscript entitled: Efficacy of single tocilizumab administration in a 88-year-old 2

patient with severe COVID-19: a case report describes a clinical case of a 88-year-old man with severe COVID-19 who was treated with tocilizumab and with a favorable outcome.

However, authors should edit the title delating the word case report because it isn’t a case report but the  description of a clinical case. I suggest this title: Efficacy of single tocilizumab administration in a 88-year-old 2 patient with severe COVID-19 and a mini literature review. Therefore authors should add a literature review

 Tocilizumab should report with capital letter also in the text

-Authors should to create a table with the laboratory values in the case description in order to summarize the case

- Authors should specify also if the patient presented COVID-19 pneumonia and should add the CT image also of the angiography CT in order to visualize the lung embolism

- Authors should divide the discussion from  the conclusion, extending also the discussion reporting other articles. I suggest to report a mini literature review and some articles as the following ones:

Al-Baadani, A., Eltayeb, N., Alsufyani, E., Albahrani, S., Basheri, S., Albayat, H., ... & Elzein, F. (2021). Efficacy of Tocilizumab in Patients with Severe COVID-19: Survival and Clinical Outcomes. Journal of infection and public health.

Duarte‐Millán, M. A., Mesa‐Plaza, N., Guerrero‐Santillán, M., Morales‐Ortega, A., Bernal‐Bello, D., Farfán‐Sedano, A. I., ... & Ruíz‐Giardín, J. M. (2021). Prognostic factors and combined use of tocilizumab and corticosteroids in a Spanish cohort of elderly COVID‐19 patients. Journal of medical virology.

Zhao, J., Cui, W., & Tian, B. P. (2020). Efficacy of tocilizumab treatment in severely ill COVID-19 patients. Critical Care24(1), 1-4.

Tsai, A., Diawara, O., Nahass, R. G., & Brunetti, L. (2020). Impact of tocilizumab administration on mortality in severe COVID-19. Scientific reports10(1), 1-7.

Author Response

Dear reviewer,

  1. The title was changed as recommended.
  2. Tocilizumab has been reported with capital letter in the text.
  3. Table 1 with laboratory values made to date was added.
  4. Figure 1 represented the thoracic radiography was added.
  5. The discussion was divided from  the conclusion.
  6.  The discussion was extended reporting other articles as suggested by reviewer.

Round 2

Reviewer 2 Report

We had requested several points, and the authors responded properly. Now the manuscript is suitable for the publication.

Author Response

Dear reviewer,

The CT scan of chest angiography is added as Figure 2.

Reviewer 3 Report

well done.

Author Response

Dear reviewer,

We add the figure of the CT scan of chest angiography.

Reviewer 4 Report

Authors should add the image of the chest CT showing the lung thromboembolism 

Author Response

(The authors gave the same response as above.)
